
# Adjustment of one-minute rain gauge time series using co-located drop size distribution and wind speed measurements

Arianna Cauteruccio [1,2], Mattia Stagnaro [1], Luca G. Lanza [1,2], Pak-Wai Chan [3]

[1]1 Department of Civil, Chemical and Environmental Engineering, University of Genova, Genoa, 16145, Italy
[2]WMO Measurement Lead Centre "B. Castelli" on Precipitation Intensity, Genoa, 16145, Italy
[3]Hong Kong Observatory, Hong Kong, China

*Correspondence to*: Arianna Cauteruccio (arianna.cauteruccio@edu.unige.it)

**Abstract.** A procedure to adjust rainfall intensity (RI) measurements to account for the wind-induced measurement bias of traditional catching-type gauges is proposed and demonstrated with application to a suitable case study. The objective is to demonstrate that adjustment curves derived from numerical simulation and disdrometer measurements allows for a-posteriori correction of rainfall time series based on the wind velocity measurements alone. One-minute RI measurements from a cylindrical tipping-bucket rain gauge installed at the Hong Kong Observatory are adjusted to quantify the impact of wind on

long-term records. Catch ratios deriving from the instrument aerodynamic behaviour under varying wind speed and drop size combinations are obtained by fitting computational fluid-dynamic simulation results already available in the literature. Co-located high-resolution wind speed measurements from a cup and vane sensor and drop size distribution measurements from an optical video disdrometer (the 2DVD) are used to infer the collection efficiency of the gauge as a function of wind speed and RI alone, and to adjust raw data from a four-year dataset (2018-2021) of one-minute RI measurements. Due to the specific

local climatology, where strong wind is often associated with intense precipitation, adjustments are limited to 4% of the RI values in 80% of the dataset. This however results in a significant amount of available freshwater resources that would be missing from the calculated hydrological and water management budget of the region should the adjustments be neglected. This work raises the need of quantifying the impact of the wind-induced bias in other sites where disdrometer data support characterizing the relationship between the drop size distribution and the measured RI. Depending on the local rain and wind

climatology the correction may account for a significant portion of the annual rainfall amount.

## 1 Introduction

The measurement bias of atmospheric precipitation gauges due to environmental sources (including wind) is strongly linked to the microphysical characteristics of precipitation (see e.g., Thériault et al., 2012), and specifically to the phase, size and fall velocity of individual hydrometeors. This link affects the efficiency of possible adjustments to the raw measured data since

they would require the measurement of additional variables and, due to the large variability of the phenomenon even at short time scales, being performed at high temporal resolution.





Dedicated sensors, called disdrometers, are generally employed to provide the particle size distribution (PSD) and fall velocity information (see e.g., Caracciolo et al., 2008). Among various applications, disdrometers can be used to improve the integral measurements of precipitation (cumulated depth and intensity) obtained from traditional gauges. Optical disdrometers (Lanza

et al., 2021) are widely used to obtain suitable PSD and fall velocity measurements. The two-dimensional video disdrometer (2DVD), manufactured by the Johanneum Research Institute, demonstrated high performance and is used in various scientific works (see Kruger and Krajewski, 2002). This specific optical sensor operates in the visible band of the light spectrum to capture images of each single falling particle that crosses the sensing volume and uses two orthogonal light sheets and two synchronized cameras to derive the 3D shape of individual hydrometeors.

The wind-induced bias of precipitation measurements obtained using catching type gauges depends – further than the instrument geometry and the wind speed – on the size of individual particles and the associated fall velocity (Leroux et al., 2021). Meickle (1819) and Jevons (1861) were the first authors to highlight the aerodynamic behaviour of precipitation measurement instruments. Evidence of the deviation of water drop trajectories close to the gauge collector due to the instrument bluff-body behaviour was initially provided by Warnik (1953) and recently quantified by Cauteruccio et al. (2021a,b) using

wind tunnel experiments.

The wind-induced bias of precipitation measurements was extensively studied in the literature and quantified using field tests (Kochendorfer et al., 2017) and computational fluid dynamic (CFD) simulations with associated particle tracking (see e.g., Něspor and Sevruk, 1999; Colli et al., 2016a,b). In the first case the ratio between the precipitation measured by a gauge in operational conditions and the reference one (obtained by properly shielding or installing the gauge) provides the so-called

collection efficiency (CE), while in the second case a numerical CE is calculated starting from the catch ratios of monodisperse precipitation later weighted and integrated after assuming a suitable PSD (see e.g., Cauteruccio et al., 2021c). The catch ratio is defined as the ratio between the number of particles captured by the gauge collector in disturbed airflow conditions and the number of particles that would be captured if the gauge was transparent to the wind.

The role of the gauge outer geometry on the aerodynamic behaviour of the gauge was investigated by Colli et al. (2018), by

simulating four different geometries: cylindrical, chimney, and two different inverted conical shapes. The CFD velocity fields, obtained within a Reynolds Average Navier Stokes (RANS) modelling approach, revealed that gauges with inverted conical shape have better aerodynamic behaviour than the chimney shaped gauge, while the cylindrical gauge had and intermediate behaviour (see also Cauteruccio et al., 2021b). The cylindrical shape is typical of most tipping-bucket rain gauges that are employed operationally. Cauteruccio and Lanza (2020), based on the CFD velocity fields provided by Colli et al. (2018),

obtained the catch ratios of a typical cylindrical gauge using the Lagrangian Particle Tracking model already validated by Cauteruccio et al. (2021a), and provided sample CE curves as a function of wind speed and rainfall intensity by using the PSD measured at an Italian test site by Caracciolo et al. (2008).

In this work, both field measurements and CFD simulation results are used to adjust liquid precipitation measurements obtained from a traditional catching-type gauge. The catch ratios provided by Cauteruccio and Lanza (2020) are expressed as a function

of both wind speed and rainfall intensity, so that the CE curves for any specific climatology can be derived based on the local





drop size distribution (DSD) and co-located rainfall intensity (RI) and wind speed measurements. A suitable formulation for the CE of a cylindrical, tipping-bucket rain gauge located in the field test site of the Hong-Kong Observatory (HKO) is derived based on a sample set of one-minute co-located RI, DSD, and wind speed measurements, starting from numerical simulation results published in Colli et al. (2018) and Cauteruccio and Lanza (2020). Adjustments are then applied to a four-year dataset

of one-minute rainfall intensity and wind speed measurements, in the absence of contemporary DSD data, and the associated measurement bias is derived. The objective is to demonstrate that adjustment curves for the wind-induced bias can be derived using CFD results and disdrometer measurements and applied for a-posteriori correction of rainfall time series based on the wind velocity measurements alone. Depending on the local rain and wind climatology the correction may account for a significant portion of the annual rainfall amount.

The organization of the paper is as follows: in the methodology section the functional dependency of the catch ratio on the particle size and wind speed is derived and disdrometer data from the investigated case study are analyzed to obtain the DSD as a function of the measured rainfall intensity. In the results section, a sigmoidal function is used to fit the numerical CE as a function of wind speed and to adjust one-minute RI measurements along a period of four years, with co-located wind speed measurements, but in the absence of contemporary DSD data. In the conclusion section results are discussed and the feasibility

and impact of the proposed methodology for the investigated case study and in other contexts is commented.

## 2 Methodology

The functional dependency of the catch ratio (CR) on the particle size (d) and wind speed ($U_{ref}$) is first made explicit for a cylindrical rain gauge. This is used to adjust one-minute rainfall intensity measurements from a tipping-bucket rain gauge installed at the HKO field test site. One-minute DSD measurements from a co-located 2DVD for a sample of rainfall intensity

events are classified as a function of RI to allow applying adjustments in the absence of contemporary DSD measurements. Then, one-minute rainfall intensity measurements from a four-year time series (2018-2021) are adjusted, and the associated measurement bias is calculated based on co-located wind speed measurements alone.

### 2.1 Functional dependency of the CR on meteorological variables

The derivation of suitable CE curves to adjust raw data obtained from a specific precipitation measurement instrument requires

the knowledge of the CRs and the local DSD. Indeed, the DSD indicates – per each drop size (usually expressed in terms of equivolumetric drop diameter) – the number of drops occurring during a precipitation event, while CRs provide the number of such drops that are not collected by the instrument in the presence of wind. In this section, we preliminarily derive the functional dependence of the CR on the drop size and wind speed for a catching-type cylindrical rain gauge.

The CR values obtained from CFD simulations with embedded particle tracking for drop diameter d from 0.25 to 8 mm and

wind speed $U_{ref}$ from 2 to 18 m s[-1] (as already provided by Cauteruccio and Lanza, 2020) are here replicated in Figure 1 with markers.



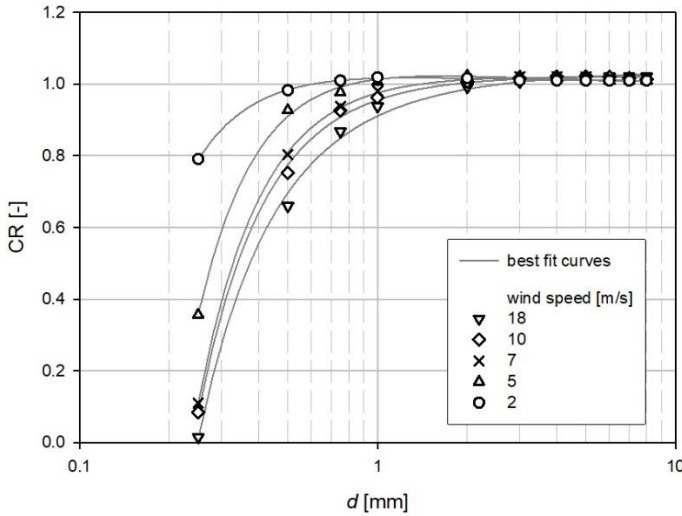

**Figure 1: Semi-logarithmic plot of the catch ratios obtained from CFD simulation by Cauteruccio and Lanza (2020) (indicated with markers) for a cylindrical gauge and best-fit inverse polynomials for selected wind speed values (continuous lines).**

The CR values collapse to unity when the drop diameter is above 2 mm, since larger drops are quite insensitive to the aerodynamic behaviour of the instruments. Note, however, that – for the case study later addressed in this work – the percentage number and volumetric fractions of precipitation for the drop size range below 2 mm are about 99.7% and 83.1%, respectively, at a RI = 1 mm h$^{-1}$, while they reduce to 98.9% and 68.0% at RI = 275 mm h$^{-1}$. The CR is therefore lower than unity for a relevant fraction of the liquid water content of the precipitation process.

To make the dependency on the drop diameter explicit, these results are here fitted (with Pearson correlation coefficients between 0.997 and 1.0) using an inverse second-order polynomial (after imposing CR = 1 at $U_{ref} = 0$) as follows:

$$CR(d, U_{ref}) = y_0(U_{ref}) + \frac{a(U_{ref})}{d} + \frac{b(U_{ref})}{d^2}, \tag{1}$$

where $y_0$, $a$ and $b$ are a function of the wind speed alone (curves are also included in Figure 1). Specifically, $y_0$ is a linear function of $U_{ref}$, while $a$ and $b$ can be expressed using a combination of a linear and an exponential function of $U_{ref}$, in the

form:

$$y_0(U_{ref}) = A \cdot U_{ref} + B, \tag{2}$$

$$a(U_{ref}) \; or \; b(U_{ref}) = C + D\,e^{-E \cdot U_{ref}} + F \cdot U_{ref}, \tag{3}$$

The constants A, B, C, D, E and F are numerical best-fit parameters. Their values are reported in Table 1, together with the associated Pearson coefficient for both the linear and mixed linear and exponential functions.


**Table 1: Numerical parameters of the functional dependence of the CR curves on wind speed (see Eq. 1).**

| Parameters of the CR curves | Parameters of the functional dependence on wind speed | | | | | | Pearson coefficient $R^2$ |
|---|---|---|---|---|---|---|---|
| | $A$ | $B$ | $C$ | $D$ | $E$ | $F$ | |
| $y_0$ | 0.0023 | 0.9985 | - | - | - | - | 0.910 |
| $a$ | - | - | 0.1106 | -0.1130 | 0.4713 | -0.0111 | 0.923 |
| $b$ | - | - | -0.1172 | 0.1195 | 0.1944 | 0.0040 | 0.962 |

Note that this formulation is site independent, since it is derived from numerical simulation results alone (by covering a wide
range of wind speed and drop size values) and can be used for most cylindrical gauges employed worldwide, provided they
have a similar outer shape than the one implemented in the simulation setup. On the contrary, the derivation of the CE curves
requires the knowledge of the DSD of precipitation events occurring under the specific climatology where rainfall
measurements are taken (a site dependent information), and therefore the use of additional local measurements. In the next
section, local measurements are used to derive the CE of the cylindrical gauge at the measurement site.

## 2.2 Field measurements

DSD measurements are obtained from a 2DVD instrument installed at the HKO field test site (see Figure 1 left-hand panel),
where the available anemometer and the tipping-bucket rain gauge are also installed. The high-quality cup and vane wind
measurement sensor manufactured by Thies Clima has a resolution of 0.1 m s$^{-1}$, while the rain gauge is manufactured by the
Shanghai Meteorological Instrument Factory (SMIF) and has a collector diameter of 200 mm and a resolution of 0.2 mm. No
filters were applied to the dataset except for selecting rainfall data at one-minute resolution with a minimum rainfall intensity
threshold equal to 5 mm h$^{-1}$.

Quality control of the data is performed by visual inspection, out of range test and jump test. The wind sensor is periodically
calibrated in the wind tunnel while the bucket volume of the rain gauge is calibrated in the laboratory once a year. The 2DVD
data are provided in the binary native form of the instrument and the DSD information is retrieved using the official software
provided by the Johanneum Research Institute.

Since rainfall measurements are provided at one-minute resolution while the DSD and wind speed measurements are available
every 10 seconds, DSD measurements are preliminarily aggregated at the one-minute resolution and a moving average is
applied to the wind measurements before sampling one-minute data. The available wind measurements are taken at the typical
height of 10 m. Although this is not optimal for the correction of the wind-induced error, wind measurements at the gauge
height are seldom available and would not be present in operational sites, therefore we propose a methodology that is easily
applicable by operational users.

DSD measurements are available for nine selected rainfall events during the years 2018-2020, for a total of about 5000 minutes
of measurements. To allow extrapolating the DSD to events with no available disdrometer information, these data are here





classified as a function of rainfall intensity (calculated from the 2DVD) and fitted within each class using the typical

exponential form (Eq. 4) as proposed by Marshall and Palmer (1948).

$$N(d) = N_0 e^{-\Lambda d} \ ,$$
(4)

with $N_0$ and $\Lambda$ being the scale and shape coefficients, respectively.

The DSD parameters can be further categorised as a function of RI using a power law function in the form:

$$\Lambda(RI) \ or \ N_0(RI) = \alpha \cdot RI^{\beta},$$
(5)

where $\alpha$ e $\beta$ are numerical best-fit parameters.

The available one-minute DSD measurements are plotted in the right-hand panel of Figure 2, where dots are color coded according to the RI measured by the 2DVD itself. Data are aggregated in classes with a diameter bin size equal to 0.5 mm and expressed in terms of number of particles (N) in a cubic meter of air per bin size. The best-fit parameters $\alpha$ e $\beta$ used in Eq. (5) are equal to 922.47 and 0.92 for the scale parameter $N_0$, and to 2.83 and -0.04 for the shape parameter $\Lambda$.

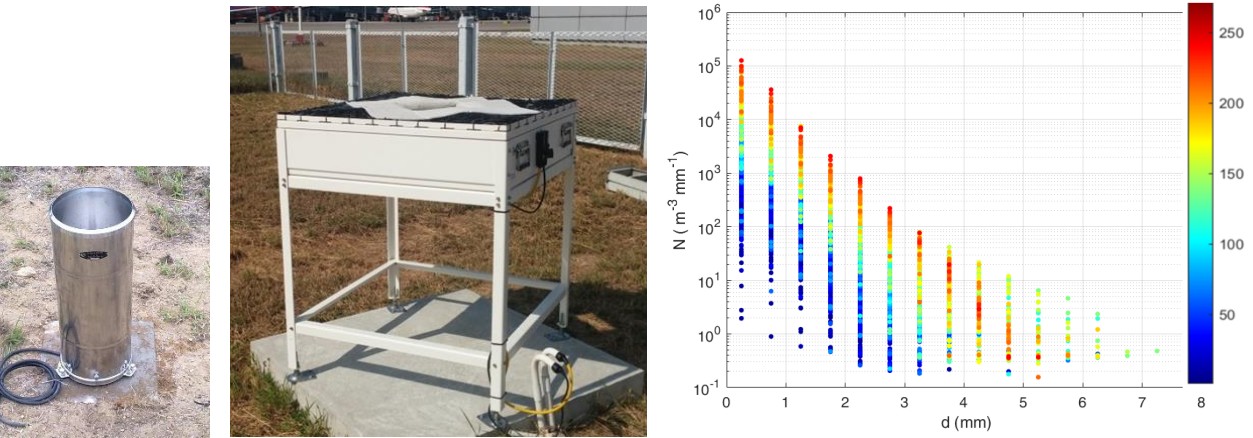


**Figure 2: The SMIF tipping-bucket rain gauge (left-hand panel) and the 2DVD instrument (central panel) installed at the HKO field test site  and DSD measurements colour coded (see the lateral colour bar) according to the rainfall intensity (RI) as measured by the 2DVD. N is the number of particles in a cubic meter of air per bin size while d is the drop diameter.**

Wind speed and RI measurements from the installed cup and vane anemometer and cylindrical tipping-bucket rain gauge are

available for a period of four years (2018-2021). In Figure 3, the variability of the measured RI with wind speed is shown for the whole period using the boxplot representation. Boxes and whiskers encompass the central 50% and 80% of the RI values, respectively, while dots indicate the 5th and 95th percentiles of their non-parametric distribution. The mean and median values are depicted with the thick and thin horizontal lines, respectively. Wind speed classes are defined using a bin width equal to 1 m s$^{-1}$. Wind speed values lower than 6 m s$^{-1}$ occur during light precipitation events (white boxes below 15 mm h$^{-1}$), while the

mean value and the variability of RI increase with increasing the wind speed and the highest RI values occur for wind speed





between 10 and 12 m s⁻¹ and at 17 m s⁻¹. Obviously, a low number of precipitation events is recorded in combination with very strong wind speed.

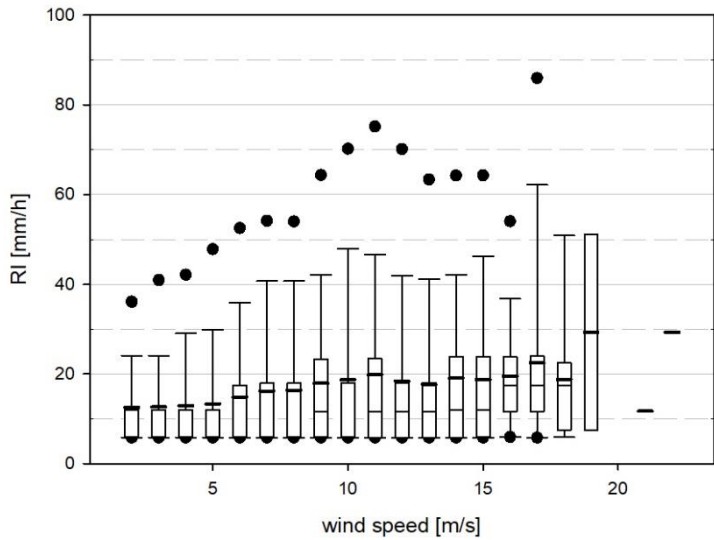

**Figure 3: Joint one-minute rainfall intensity (RI) and wind speed climatology at the HKO field test site for wind speed bins of 1 m s⁻¹. Boxes and whiskers encompass the central 50% and 80% of the sample, respectively, while dots indicate the 5ᵗʰ and 95ᵗʰ percentiles of their non-parametric distribution. The mean and median values are depicted with the thick and thin horizontal lines, respectively.**

**3 Formulation of the CE and data adjustment results**

The functional dependency of the CR derived as in Eq. (1), with the associated parameters expressed as a function of wind speed, is used in association with the exponential DSD, with parameters expressed as a function of RI (as obtained from the sample of selected rainfall periods), to numerically calculate the CE of the tipping-bucket rain gauge for use when no direct disdrometer measurements are available.

The calculation is performed for sampled RI values equal to 1, 3, 5, 10, 15 mm h⁻¹ and from 25 to 275 mm h⁻¹ at increments of 25 mm h⁻¹. A sigmoidal function is used to fit the numerical CE as a function of wind speed, in the form:

$$CE(U_{ref}) = \varepsilon_0(RI) + \frac{m(RI)}{1+e^{\frac{\left(U_{ref}-x_0(RI)\right)}{n(RI)}}}, \tag{6}$$

with $\varepsilon_0$, $m$, $x_0$, and $n$ being the best-fit parameters reflecting the dependency of the CE on RI.

A visual representation of the above exercise is reported in Figure 4, for about 700 minutes of recorded precipitation with wind speed larger than 2 m s⁻¹, where each grey circle represents a one-minute numerical CE value. The size of each circle provides an indication of the associated RI value, with thin circles indicating light RI and exhibiting lower CE values at any wind speed





than in the case of intense RI (large circles). Overlapped are the CE curves calculated for a few sample RI levels, including

the maximum and minimum RI to enclose the entire range of measured data, and plotted over the whole wind speed range.

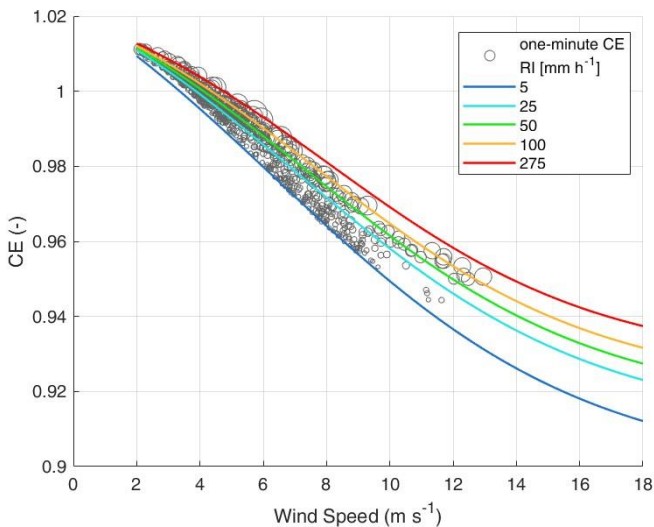

**Figure 4: Collection Efficiency (CE) values (grey circles) as derived from one-minute DSD and wind speed measurements with overlapped the CE curves (solid lines) at fixed rainfall intensity (RI) values and for the entire wind speed range. The size of each**
**circle provides an indication of the magnitude of the associated RI value.**

This functional dependency on RI can be described, for all sigmoidal parameters, by a logarithmic function expressed as:

$$f(RI) = G + H \cdot \ln(RI), \tag{7}$$

where $f(RI)$ assumes the role of the functional dependency of the $\varepsilon_0$, $m$, $x_0$, and $n$ parameters.

**Table 2: Numerical constants of the logarithmic functional dependency of the four parameters of the sigmoidal CE curve on RI and**
**the associated correlation coefficient ($R^2$).**

| Parameters of the sigmoidal CE | Parameters of the functional dependency on RI | | Pearson correlation coefficient $R^2$ |
|---|---|---|---|
| | $G$ | $H$ | |
| $\varepsilon_0$ | 1.0654 | -0.0058 | 0.984 |
| $m$ | -0.1792 | 0.0132 | 0.994 |
| $x_0$ | 5.6607 | 0.4366 | 0.997 |
| $n$ | -5.1739 | 0.1371 | 0.997 |

The best fit of the CE parameters as a function of RI is shown in the panels of Figure 5. An optimal trend with RI was observed since the Pearson coefficient ($R^2$) assumes the minimum value equal to 0.984 in the case of $\varepsilon_0$.



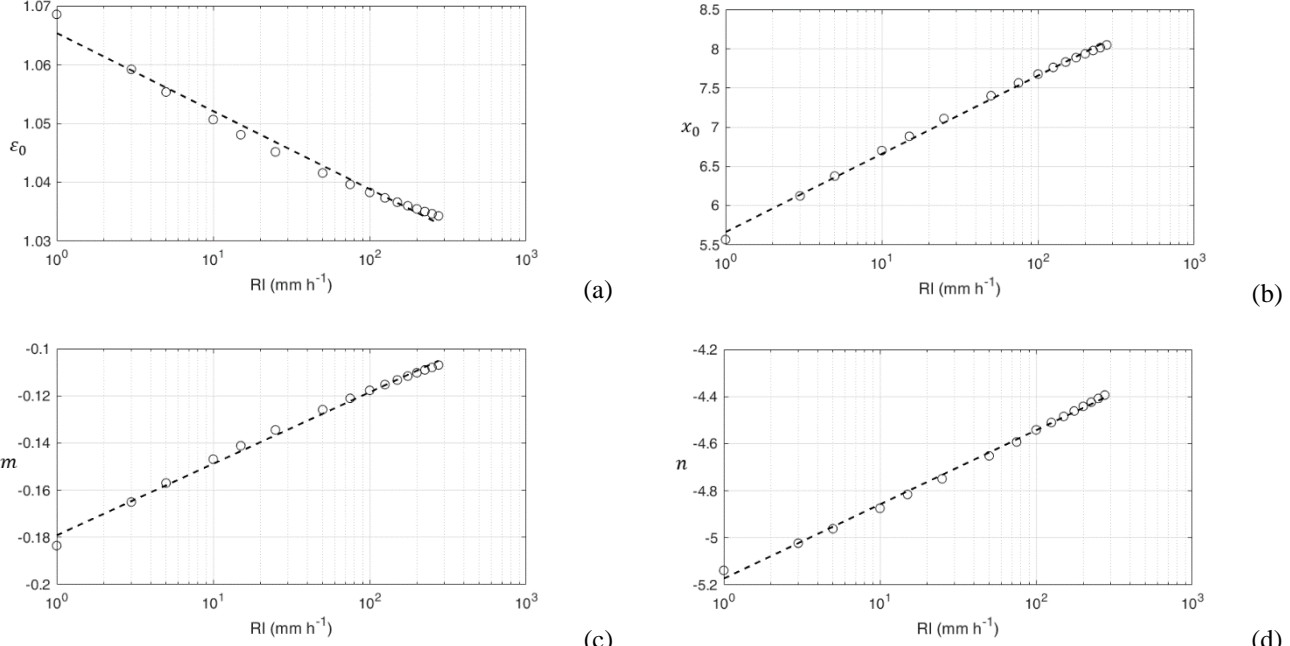

**Figure 5: Logarithmic best-fit functional dependency of the sigmoidal Collection Efficiency (CE) parameters ($\varepsilon_0$, $x_0$, $m$ and $n$) – see Eq. 6 – on the rainfall intensity (RI).**

The derived CE formulation is used to adjust one-minute RI measurements along a period of four years (2018-2021), with co-located wind speed measurements, but in the absence of contemporary DSD data.

Results of the adjustment for the whole period, expressed in terms of the applied CE, are visualised in Figure 6 as a function of wind speed. One-minute CE values (reported on the left-hand axis) are summarized in the form of boxplots. Boxes and whiskers encompass the central 50% and 80% of the CE values, respectively, while dots indicate the 5[th] and 95[th] percentiles of their non-parametric distribution. The mean and median values are depicted with the thick and thin horizontal lines, respectively. Wind speed classes are used with a bin width equal to 1 m s$^{-1}$. The grey histogram indicates the sample size (quantified on the right-hand axis) for each wind speed class. Results show that predominant wind speeds are between 3 and 5 m s$^{-1}$. The variability of the CE values, for each wind speed class, is coherent with the variability of the RI (illustrated in Figure 3) and is not ascribable to the sample size.





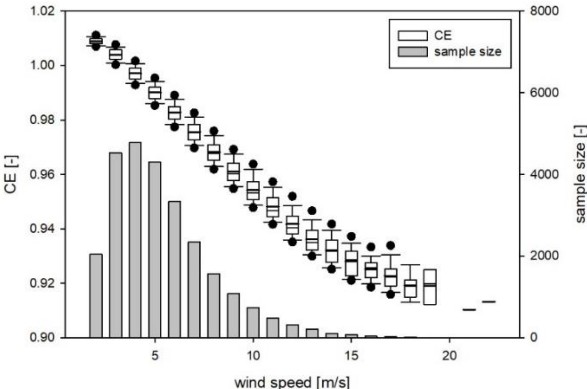

**Figure 6: One-minute Collection Efficiency (CE) values (left-hand axis) for wind speed classes with a bin width equal to 1 m s⁻¹. Boxes and whiskers encompass the central 50% and 80% of the dataset, respectively, while dots indicate the 5th and 95th percentiles of their non-parametric distribution. The mean and median values are depicted with the thick and thin horizontal lines, respectively. Grey bars indicate the sample size for each wind speed class (values on the right-hand axis).**

For each minute, the complement to unity of the CE i.e., the percentage relative bias is also calculated in the form:

$$e(\%) = \frac{RI_{corr} - RI_{meas}}{RI_{corr}} \cdot 100, \tag{8}$$

where $RI_{meas}$ is the measured RI value, while $RI_{corr}$ is the same variable after the adjustment.

In Figure 7, the calculated relative bias [%] is visualized in the form of boxplots for the four investigated years individually, and compared with the associated non-parametric RI and wind speed distributions. Results reveal that most relative biases are lower than 3%, whereas the maximum values reach about 10%. The dependency of the relative bias on the wind speed is predominant since the occurrence of higher rainfall intensity values in 2020 and 2021 does not result in a reduction of the relative bias.

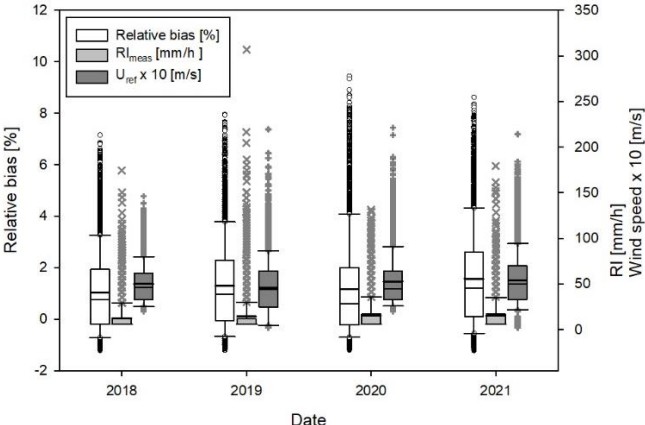

**Figure 7: Non-parametric distribution of the one-minute percentage relative bias [%] (left-hand axis), measured rainfall intensity (RI_meas) and wind speed (U_ref) (right-hand axis) for the four investigated years individually. Boxes and whiskers encompass the central 50% and 80% of the dataset, respectively, while markers indicate values out of the 5th and 95th percentiles of their non-parametric distribution. The mean and median values are depicted with the thick and thin horizontal lines, respectively.**





## 4 Conclusions

To overcome the wind-induced bias of traditional catching-type rain gauges a suitable formulation of the CR as a function of the drop diameter (and the dependency of the parameters of such formulation on the wind speed) can be used to adjust precipitation intensity measurements from any specific instrument. Knowledge of the local DSD dependency on the RI, as obtained from a reference disdrometer, and high-resolution wind measurements are needed to derive dedicated CE functions that may be applied to adjust the raw measurements for any location and rain event characteristics. The proposed procedure is

replicable in each site where contemporary rainfall intensity, wind speed and DSD measurements are available, while the CE curves derived for the field test site of Hong Kong are applicable in all sites having the same rainfall climatology.

Note that the reference disdrometer is subject to wind-induced biases itself, as demonstrated by Něspor et al (2000) for a now outdated version of the 2DVD and by Chinchella et al. (2021) for the Thies LPM optical disdrometer. Further insights could be therefore achieved by obtaining bias corrected DSD measurements, a topic which is not addressed in this paper. Once the

wind-induced bias on DSD measurements from a 2DVD is quantified, corrected DSD measurements should be used to derive updated CE curves following the procedure proposed in the present work.

The mathematical formulation of the CR as a function of the drop diameter and the wind speed is obtained in this work by fitting numerical fluid-dynamic simulation results already available in the literature. The direct dependency of the sigmoidal form of the CE on the wind speed, and the derived experimental relationship between its parameters and RI, streamline its

practical application.

The use of the proposed method to apply corrections for the wind-induced bias of catching-type gauges is therefore straightforward and requires no further information than the measured rainfall intensity and the wind velocity (with no need of having a disdrometer operating at each measurement site). The described application to a four-year dataset of co-located precipitation intensity and wind measurements at the resolution of one minute demonstrates the feasibility of the proposed

adjustment methodology.

Depending on the local rain and wind climatology the correction may account for a significant portion of the annual rainfall amount. At the location chosen for the presented case study, due to the specific local climatology where strong wind is often associated with intense precipitation (dominated in volume by large drops with limited susceptibility to the aerodynamic behavior of the instrument), adjustments are limited to 4% of the one-minute precipitation intensity in 80% of the dataset. This

results on average in about 1.32% of the measured yearly precipitation (totalizing about 1550 mm), a small percentage that however sums up to $23 \cdot 10^6$ cubic meters of fresh water if computed over the eleven hundreds square kilometers territory (land area only) of the Hong Kong special administrative region of the People's Republic of China. The adjustment therefore accounts for about the storage capacity of the local Shek Pik reservoir, the third in terms of capacity within the Hong Kong water supply system. Such significant amount of available freshwater resources would be missing from the calculated

hydrological and water management budget of the region should the adjustments for the wind-induced bias of traditional precipitation measurements be neglected.



Adjustments calculated at other locations having different precipitation and wind climatology than the presented case study may show different results, and further work is ongoing to compare the impact of the wind-induced bias in various sites where disdrometer data support characterizing the relationship between the DSD and RI.

**Data availability**

The data used in this work are available on request from the corresponding author.

**Author contribution**

AC and LGL developed the methodology and conceptualisation, PWC selected and provided the data, AC and MS performed the investigation and data analysis and visualisation, AC wrote the original draft, LGL reviewed and edited the manuscript and supervised the work. All authors have read and agreed to the published version of the manuscript.

**Competing interests**

The authors declare no conflict of interest.

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
