# Peer review of "Adjustment of one-minute rain gauge time series using co-located drop size distribution and wind speed measurements"

_Atmospheric Measurement Techniques, 2023_

## Author Response (AR1)

**Responses to RC1**

The study focuses on the adjustment of rainfall measurements using wind speed and particle size distributions. It uses mutli-year data collected at a site where a precipitation gauge, a wind speed sensor and a disdrometer (2DVD) were installed together. Computational fluid dynamics simulations are used to obtain a theoretical curve for the catch efficiency. The manuscript is well written and present new findings in the field of rainfall measurements. I only have a few specific questions/comments to improve clarity.

Specific comments:

1. Line 21: 'however' should be between commas. Please correct throughout the manuscript.

   Done.

2. Lines 19-21: The sentence: "Due to the specific local climatology, where strong wind is often associated with intense precipitation, adjustments are limited to 4% of the RI values in 80% of the dataset." What do you mean by "adjustments are limited to 4% of the RI values in 80% of the dataset"? Do you mean that it only increases precipitation amounts by 4% and it only impacts 80% of the dataset? Please clarify this sentence. It seemed to be an important as the same information is repeated in the conclusion (L254).

   The statement means that for most of the investigated minutes (about 80% of the whole period as indicated by the boxplot whiskers in Figure 7), the adjustment is within 4% of the observed RI. This value depends on the climatology of the site, as mentioned in the paper. In the presented case study the value is not large but, as explained in the conclusions, it sums up to a yearly water volume equivalent to the capacity of the third largest dam within the Hong Kong water supply system, when applied to the whole territory of Hong Kong.
   We modified the sentence into "in 80% of the dataset adjustments are limited to 4% of the observed RI values", to make it clearer.

3. Both 'collection efficiency' and 'catch ratio' are used in the manuscript. I suggest that the authors only use either 'CR' or 'CE' in the text for clarity.

   The catch ratio (CR) and the collection efficiency (CE) are two different variables, and they are both used (and necessary) in the paper to obtain the adjustments. The catch ratio (CR) represents the impact of wind on the instrument external geometry for a monodisperse rainfall with a given raindrop size. It is a function of the wind speed and the instrument geometry, while it is independent on the location of the measurement site and the instantaneous rainfall intensity. The collection efficiency (CE) is later derived from the catch ratios by assuming a given drop size distribution (DSD), which is linked to the rainfall intensity. This allows obtaining the CE as a function of the wind speed and rainfall intensity alone. The obtained CE is therefore site-dependent since it derives from the local rainfall climatology (e.g., the predominance of either convective or stratiform rain), through the associated DSD. (see L89-93)
   We believe that both variables are essential to the proposed procedure and should therefore be included in the manuscript.

4. Lines 98-99: 1) In the figure 1 caption, the authors should add the variable used on the y-axis in parenthesis after 'catch ratios'. 2) The figure caption seems incomplete. Add 'as a function of diameter (d)' at the end of the sentence.

Done.

5. All Figures: The author should add letters to the panels and refer to the letter. It is much easier to follow other than using left-hand, center, and right-hand panels in the caption and the text.

Done.

6. Lines 126 and 142: The sentence should start with 'The'.

Done.

7. Line 144: Equation 4 is an inverse exponential. Add 'inverse' before exponential.

Done.

8. Lines 142-147: Why is it needed to estimate the missing particle size distribution using the observed rainfall intensity? Also, can you elaborate on the need to have equation 5? I am asking because Marshall and Palmer (1948) propose an equation that relates the slope of the size distribution and the precipitation rate. They also showed that No is constant for rainfall. How does equation 5 compares with the Marshall and Palmer relationship? Additional explaination should be added to the manuscript.

The rainfall intensity is a function of the drop size distribution (DSD). Linking the DSD with the rainfall intensity is the method adopted here to derive the parameters of such function when no disdrometers are available at the site for direct measurement. To ensure that the procedure proposed in this work can be exported to any measurement site where only a catching-type gauge is used (like most of the operational weather stations), we used the observed rainfall intensity as a surrogate for the DSD.
In our study, however, direct DSD measurements were also available, although for a limited period. Therefore, a relationship between the rainfall intensity and the DSD could be derived for use throughout the whole dataset. Equation 5 is the same equation proposed by Marshall and Palmer (MP) to relate the parameters of the exponential DSD with the rainfall intensity, but with site-specific parameters (MP suggest the values of $\alpha$ and $\beta$ for the coefficient $\Lambda$ and $N_0$ = cost). Indeed, it was shown in the literature that the MP formulation is not universal and that $N_0$ is not necessarily constant (as is also evident from the data reported in Figure 2). Obviously, the relationship proposed by MP could be used as well, in the absence of a more specific determination of the parameters of Equation 5, as a suitable approximation.
Additional explanations have been added to the manuscript to clarify these aspects before section 2.2.

9. Lines 201-202: A paragraph should be more than one sentence. You could elaborate more or merge with previous/next paragraph.

Done.

10. Lines 203-209: Most of the text should be move to the figure caption.

We added the information about the bin size in the caption and we removed the repetition in the text.

11. Lines 219-220: Most of the text should be move to the figure caption.

The relevant information is already included in the caption of Figure 7. We prefer to maintain the information also in the text.

**Responses to RC2**

The study focuses on the adjustment of rainfall measurements using wind speed and rain size distributions from a disdrometer (2DVD). Catch efficiency is calculated for different rates and wind speeds using computation fluid modeling. Overall the manuscript is well written butI do have a few questions/comments to improve the manuscript.

**Major Comments:**

CR and CE seem to be used interchangeably throughout the paper. I would suggest standardizing on one.

The catch ratio (CR) and the collection efficiency (CE) are two different variables, and they are both used (and necessary) in the paper to obtain the adjustments. The catch ratio (CR) represents the impact of wind on the instrument external geometry for a monodisperse rainfall with a given raindrop size. It is a function of the wind speed and the instrument geometry, while it is independent on the location of the measurement site and the instantaneous rainfall intensity. The collection efficiency (CE) is later derived from the catch ratios by assuming a given drop size distribution (DSD), which is linked to the rainfall intensity. This allows obtaining the CE as a function of the wind speed and rainfall intensity alone. The obtained CE is therefore site-dependent since it derives from the local rainfall climatology (e.g., the predominance of either convective or stratiform rain), through the associated DSD (see L50-54 in the revised manuscript).
We believe that both variables are essential to the proposed procedure and should therefore be included in the manuscript.

The paper could use a little bit of reorganizing to introduce some of the sensor information earlier so the reader isn't left to guess what the authors are referring to. A map of the site location would also be useful and some images showing not just the sensors, but where they were placed in relation to each other at the site (this is almost mentioned below in regards to line 84).

We have provided the requested information earlier in the manuscript by reorganizing the whole section 2 and anticipating sub-section 2.2 before the original 2.1.
A map of the site location and a picture of the installed sensors were added as a new Figure 1.

**Minor comments:**

Line 32: Employed should be either "used" or "deployed".

Done.

Line 36: the word "has" should be inserted before demonstrated.

Done.

Line 43-45: Was the gauge(s) used in this study the same as the ones described in the cited literature? If not, do the authors feel that the results are applicable to this study?

Yes, we have added the following sentence to better clarify: "where various gauge geometries, including the cylindrical shape, were tested".

Line 57: The "and" near the end needs to be an "an".

Done.

Methodology Intro (lines 82-87): The authors need to state the make and model of the gauge they are using or provide the dimensions of the modeled gauge. It's confusing the reader here to not have that information at this point.

We added the gauge manufacturer and, while reorganizing the section, we moved the information about the involved instruments earlier, at the beginning of Section 2.

Line 84: A photo of the sensors at the site would be useful to reference here (similar to Figure 2 but showing more of the site so any nearby obstacles or terrain can be shown).

Done.

Line 126: There is no left-hand panel in figure 1, I assume the authors are referring to Figure 2 here.

Yes, fixed, thanks (also considering the request of the first reviewer to change panel positions with lettering).

Line 129: The information on the gauge here should really come much earlier in the paper, otherwise the reader has no reference as to what the authors are talking about until this point. Also, information on the maximum measurable rate of the precipitation gauge would be useful to be included here.

Done.

Line 154: what is the "e" referring to between alpha and beta? I'm also a bit confused on what exactly the -2.83 and -0.04 numbers are referring to.

The world "e" was replaced with "and". The values -2.83 and -0.04 refer to the best-fit coefficients ($\alpha$ and $\beta$) of Eq. 5 for the shape parameter $\Lambda$. To improve clarity, we changed "parameters" with "coefficients" when referring to $\alpha$ and $\beta$, while $N_0$ and $\Lambda$ are now called "parameters". This nomenclature was made consistent throughout the manuscript.

Figures 3, 6, and 7: The authors use m s$^{-1}$ (for example) in the caption but use m/s in the axis labels of the plot. I would suggest changing the plot axis labels to match the rest of the paper for consistency.

Done.

Line 184: "larger" should be "higher".

Done.

Figure 4: The legend on this figure is very confusing. As I read it, the legend says the circles are the one-minute CE RI [mm h$^{-1}$] (which doesn't make sense) but the caption indicates they are the

magnitude of the Ri values. I would suggest redoing the legend and splitting it into two legends; one that gives the line definitions and one that gives a scale to better indicate the magnitude of the circles being shown. The size difference in the circles is hard to see with so many overlapping circles. The authors could consider making the lines one color and using dashes and dots to distinguish them and color-coding the circles instead. This would help should the larger circles have a higher CE. The first sentence in the caption also doesn't make sense and needs to be reworded (I believe the "the" before CE curves needs to be removed).

Circles indicate the one-minute CE while RI [mm h$^{-1}$] refers to the RI values of the solid lines. We have modified the legend in order to avoid any misunderstanding.

About the colours of dots and lines, we have checked the suggested solution and we think that the original version is clearer and more understandable. Note that each dot positioned between two lines have an intermediate RI value between those indicated by the line colours.

We removed the article "the" before CE in the first sentence of the caption.

Line 235: Do you have a reference to the climatology for the rainfall at HKO?

We added reference to two papers dealing with the rainfall climatology at Hong Kong.

Lines 254-257: The grammar needs to be fixed in this sentence.

The sentence was revised by checking grammar and separating it in two sentences.

Line 262: I'm not sure the authors have shown an actual precipitation and wind climatology in this paper (or provided a reference to one for this site). Without that, how would others know if this information can be used for gauges at their sites?

We did not show the precipitation and wind climatology at the study site but reported in Figure 7 the one-minute rain and wind data distribution observed in four consecutive years. We then provide an example of the impact of neglecting the wind-induced bias on a series of observations spanning over four years (where high-resolution rain gauge and 2DVD data were available). This is sufficient to show that the expected impact is significant.

The actual climatology at the site of interest does not affect the proposed methodology but (in case of any different precipitation and wind climatology) a higher or lower impact can be expected depending on whether light to moderate rainfall intensity events occur with strong wind or not.

In Hong-Kong, strong wind is common during high intensity precipitation (see Figure 7), therefore – as explained in the manuscript – the impact of the wind-induced bias is dampened because large drops are less sensitive to the wind. In case strong wind is accompanied by low to moderate rainfall intensity events, as it may be the case elsewhere, the impact of the wind-induced bias is expected to be much higher.

The methodology, rather than the specific information provided in the paper, is not site dependent and can be used at any site where contemporary rainfall intensity, wind speed and DSD measurements are available.

---

## Referee Report (RR1)

The authors have done a nice job of addressing the reviewer's comments in the first round of reviews. I do have a few additional things I would like to see the authors address.

1) If the authors are going to keep CE and CF, they need to define each of them in the paper. Many in the precipitation measurement community consider them to be the same so if the authors wish to define them differently, they need to do so in the manuscript.
2) I believe your figure numbers are off in the manuscript (there is no figure 3 for example).
3) Your equation numbers similarly need to be revisited (your first equation shows up as equation 14 now instead of equation 1).
4) The addition of Figure 1 is excellent. My only suggestion is that the arrows on 1b are so thin, it's hard to see where they are pointing to. I'd suggest making them thicker and/or changing the color.
5) In Figure 1b, it might be useful to also depict the wind direction most commonly associated with the precipitation. I'm assuming it's from the bottom of the picture towards the top since the shorter gauges are located near the bottom of the picture? It would help alleviate any questions about wind impacts from the taller sensors on the shorter sensors.

---

## Author Response (AR2)

Report#1:
Thank you for answering all my questions and comments. In particular, I think that it was important to define the difference between the catch ratio and the collection efficiency for clarity. Finally, I would suggest to delete the word 'obviously' on line 134.

Thanks, we have removed the word 'obviously' as suggested.
* * *
Report #2:
The authors have done a nice job of addressing the reviewer's comments in the first round of reviews. I do have a few additional things I would like to see the authors address.

1) If the authors are going to keep CE and CF, they need to define each of them in the paper. Many in the precipitation measurement community consider them to be the same so if the authors wish to define them differently, they need to do so in the manuscript.

The variables CE and CR are defined at lines 48-49 and 52-53, respectively.

2) I believe your figure numbers are off in the manuscript (there is no figure 3 for example).

I correctly see Figure 3 in my word manuscript, the problem is due to the generation of the pdf file. To avoid this problem, I've removed the automatically renumbering of figures.

3) Your equation numbers similarly need to be revisited (your first equation shows up as equation 14 now instead of equation 1).

The first equation is correctly labelled as Equation 1. The number 4 is barred because the equation was moved after the inversion of sub-sections 2.2 and 2.1, as requested during the first round of review.

4) The addition of Figure 1 is excellent. My only suggestion is that the arrows on 1b are so thin, it's hard to see where they are pointing to. I'd suggest making them thicker and/or changing the color.

Thanks, the figure was modified accordingly.

5) In Figure 1b, it might be useful to also depict the wind direction most commonly associated with the precipitation. I'm assuming it's from the bottom of the picture towards the top since the shorter gauges are located near the bottom of the picture? It would help alleviate any questions about wind impacts from the taller sensors on the shorter sensors.

A white arrow indicating the prevailing wind direction (about 100 degrees) is now included in Figure 1b. Therefore, wind mostly impacts from the lower to the taller sensors.
* * *
Note from review file validation:
For the next revision, please check the figure labels ("Figure 1" appears twice)

I correctly see Figure 1 in my word manuscript, the problem is due to the generation of the pdf file. To avoid this problem, I've removed the automatically renumbering of figures.

---

## Author Response (AR3)

One small thing in Figure 1b: As you mention in the text, the prevailing wind direction is about 100° but the white arrow in the Figure looks more like 140-150°. Please check for correctness.

Since Figure 1b is a photograph, the direction of the white arrow (the prevailing wind direction) is just indicative. However, to be more precise, we modified both Figures 1a and 1b to include the north direction and (in Figure 1b) a white arrow at about 100° indicating the wind direction.